# Study on Crust-Shaped Dust Suppressant in Non-Disturbance Area of Open-Pit Coal Mine—A Case Study

**DOI:** 10.3390/ijerph20020934

**Published:** 2023-01-04

**Authors:** Zhiyuan Shen, Zhongchen Ao, Zhiming Wang, Yuqing Yang

**Affiliations:** 1Inner Mongolia Bureau of the State Mine Safety Supervision Bureau, Hohhot 010010, China; 2School of Mines, China University of Mining and Technology, Xuzhou 221116, China

**Keywords:** open-pit coal mine, dust pollution, dust particle size, dust suppressant, dust suppression test

## Abstract

Dust pollution in open-pit coal mines severely restricts the green development of mines. Therefore, dust control has become an important requirement for the sustainable development of the mining industry. With the goal of dust pollution prevention and control in open-pit coal mines, this paper puts forward the concept of a non-disturbance area of an open-pit coal mine. It clarifies the characteristics of dust generation, the coverage area, and the dust particle size distribution characteristics of the non-disturbance area. Taking the dust control at the dump site as an example, the study comprehensively utilizes indoor tests and field tests to develop a dust suppressant for the dump site and determine its dust suppression efficiency and effective service cycle. The results show that the D10, D50, and D90 particle sizes of dust in the non-disturbance area are smaller than those in the disturbance area, and the difference in particle size of D90 is the most obvious. Gelatinized starch and non-ionic polyacrylamide, as the main components of the dust suppressant, can effectively reduce dust pollution in the dump; the optimal concentration is 1.0%, and the dust suppression service cycle is more than one month. The developed dust suppressant does not contain corrosive, toxic, or heavy metal elements. Although the application of a dust suppressant will cause plant growth to lag, it does not affect plant health. The research findings serve as a reference for the zoning treatment of dust in open-pit mines.

## 1. Introduction

The basic national conditions of China’s energy distribution are “rich coal, poor oil, and low gas” [1,2]. According to statistics, coal energy consumption accounted for 56% of China’s total energy consumption in 2021, which shows that coal is still a vital energy pillar of China’s national economic development. Coal mining has ensured the rapid development of China’s economy. However, it has also caused environmental problems, such as dust pollution [3], terrain collapse, noise pollution, and vegetation destruction. Under the comprehensive influence of global environmental changes and the international situation, the green mining of coal has become the only way to sustain the coal industry [4,5]. Open-pit mining is the predominant technique for extracting solid minerals because of its safety and productivity [6,7,8]. However, a large amount of dust and exhaust gas produced by open-pit mining is directly exposed to the atmosphere, which leads to the severe problem of particulate pollution. Open-pit mining dust pollution presents the characteristics of “many points, wide areas, and high pollution concentrations”, involving multiple operations such as perforation, blasting, collection, transportation, discharge, and unloading. Dust not only pollutes the mining area and the surrounding environment [9] but also endangers human health and causes occupational diseases such as pneumoconiosis and silicosis. With the depletion of coal resources in Central and Eastern China, the focus of coal mining has gradually shifted westward, and the coal production capacity in Xinjiang, Inner Mongolia, and other areas has gradually increased. This speed has accelerated significantly in recent years. Xinjiang, Inner Mongolia, and other areas are mainly located in cold, arid, semi-arid, and ecologically fragile areas, with little precipitation, extensive evaporation, and weak environmental self-purification ability. Therefore, dust pollution problems are further highlighted in these areas [10].

Many scholars have researched the dust initiation, diffusion, monitoring, and prediction of open-pit dust. Among these, Li and Zhang [11] focused on monitoring the dust concentration in and around the Antaipo open-pit coal mine and conducted separate dust sampling for the types of workers exposed to high dust concentrations. AERMOD software (Version 12345, Washington, DC, USA) was used to simulate dust diffusion, establish a dust concentration evaluation and diffusion model, and obtain a contour map of the dust concentration around coal mines. Huertas et al. [12] collected and analyzed topographic and meteorological information in the area based on readings reported by the Northern Colombian Coal Mine Air Quality Monitoring Network, using meteorological data collected by three local stations in 2008 and 2009 to simulate the dispersion of TSP in ISC3 and AERMOD. Comparing the obtained results with the actual values measured by the air quality monitoring network yielded a high correlation coefficient (>0.73), indicating that the model accurately describes the main factors affecting particle dispersion in this area. Ding et al. [13] investigated the wetting performance of several commonly used surfactants on coal dust and their properties after magnetization and experimentally found that the surface tension of surfactants decreased with the increase in their concentration. Xi et al. [14] mixed polyethylene glycol with sodium lauryl sulfate to obtain a dust suppressant and found that the surface of coal dust covered by the dust suppressant formed a protective film layer, which could maintain the effect of suppressing dust for a long time. The inhibition effect of water sprinkling often depends on the physicochemical properties of the particle surface. The effect of spraying pure water on hydrophobic dust is very poor. The addition of surfactants to water can reduce the surface tension of water and enhance the wettability of particles [15]. Researchers prepared a biological dust suppression agent using urease extracted from soybean to evaluate the application of enzyme-induced carbonate precipitation technology in controlling coal dust pollution in open-pit coal mines [16,17]. Jin et al. [18] used the anionic surfactant sodium lauryl sulfate to modify a natural, biodegradable soybean protein isolate as a dust suppressant to reduce dust diffusion pollution in open-pit coal mines. Medeiros et al. [19] used the glycerol by-product of biodiesel to prepare a highly effective dust suppressant at 140 °C, which can effectively prevent fugitive particulate matter emissions. Yan et al. [20] explored the performance of modified sodium alginate as a coal dust inhibitor. The results showed that the material had good water retention, wettability, and shell formation for coal dust. Its spray dust suppression efficiency could reach approximately 90%. Dixon-Hardy et al. [21], of the University of Leeds in the United Kingdom, explored the possibility of preparing mine dust suppressants from refinery waste, and the study showed promising results.

There are many dust suppressants for open-pit mines, and the types of dust suppressants applied in different areas are different. The previous research is mainly concentrated in open-pit mining areas. However, less research has been conducted on dust suppressants in non-disturbed environments, since the dust in these locations is fine and requires targeted treatment. In order to reduce the fugitive emissions of fine particles from open-pit lignite mines, Zhou et al. [22] proposed a dust inhibitor that included a mixture of 0.7% water-soluble polymer and 0.1% nonionic surfactant. Schwendeman et al. [23] prepared a foam dust suppressant composed of water-soluble polymer substances and surfactants for the respiratory dust generated during coal cutting and mining, with a dust suppression efficiency of 90%~95%. Li et al. [24] studied the effects of different surfactants on the wettability of pulverized coal. The results showed that the finer the pulverized coal particle size, the more complex the microstructure of coal particles, and the worse the wetting ability. The experimental study of Zhuang and Biswas [25] found that submicron particulate matter is formed by evaporation–absorption–condensation during coal combustion. The nucleation mode of ash particles can be changed by the injection of a vapor phase adsorbent. Zhou et al. [26] explored the inhibitory effect of a dust suppressant on blasting dust in a tunneling roadway by adding an SDBS surfactant and NaAc compound to the spray system, and the results showed that after adding the dust suppressant, the respirable dust concentration was reduced by 75.15% and the total dust concentration was reduced by 78.32%. Bao et al. [27] used the polymer methylene succinic acid–acrylic acid and bentonite to chemically synthesize a bonded coal dust inhibitor with moisturizing and wind erosion resistance. The dust suppressant was twice as wettable as clean water. Wang et al. [28] explored the effect of water-soluble polymers on the properties of foam dust suppressants and found that when the polymer concentration is low, it has a lesser effect on the wettability of the foam, but it can significantly improve the surface viscosity and stability of the foam, which helps to improve the dust suppression efficiency of the foam.

Different from previous studies, this paper proposes the concepts of an open-pit mine dust disturbance area and non-disturbance area according to the dust generation characteristics of an open-pit coal mine and points out that zonal dust reduction is an effective method to reduce open-pit coal mine dust pollution. A super-large open-pit coal mine in China, the Haerwusu open-pit coal mine, with an annual output of more than 35 Mt, was selected as the test base. This mine was chosen because it is located in a typical cold area where more than 90% of China’s open-pit coal production occurs. In winter, dust pollution in the stope is severe. The mining area has prolonged cold periods with low precipitation and high evaporation. It is difficult to spray water during winter, and other problems need to be solved urgently. In this paper, the difference in dust particle size distribution was obtained through indoor experiments. The dust reduction efficiency, viscosity, crusting time and thickness, toxicity, corrosion, plant growth, and heavy metal content of different concentrations of dust suppressants were analyzed. The material and ratio range of the binder in the non-disturbance zone were determined. Finally, field tests were conducted to determine the optimal ratio and service cycle of the adhesive dust suppressant in the non-disturbance zone.

## 2. Materials and Methods

The research framework of this paper is shown in Figure 1. Firstly, the dust characteristics of the open-pit coal mine operation link were analyzed, the open-pit mine was divided into disturbance areas and non-disturbance areas according to the dust characteristics, and the dust particle size distribution in different areas was given through particle size analysis. The types of dust suppressants in the non-disturbance area were determined by analyzing the dust suppression principles of different types of dust suppressants. Then, the dust suppression efficiency of different concentrations of dust suppressants was studied by wind tunnel experiments. The viscosity, crusting time, crusting thickness, and environmental impact of different concentrations of dust suppressants were analyzed through indoor tests. Finally, according to the laboratory test results, the field test was carried out to analyze the dust suppression efficiency and adequate service time and determine the optimal ratio of dust suppressant.

### 2.1. Non-Perturbation Area Definition

Coal open-pit mining is essentially a complex and orderly evolution process, from large-scale soil and rock cube crushing and transportation of the structural space to the spatial structure of the discarded and reshaped entity, involving multiple links, such as perforation, blasting, mining, transportation, crushing, and disposal. In each production process, under the influence of high-strength physical damage and disturbance, the stope dust presents significant characteristics, such as many dust sources, large dust production, and an unfixed diffusion mode and direction. The dust in the open-pit mining operation link is mostly mechanical dusting. The dust is mostly a large-particle-size primary dust source near the mechanical equipment concentrated in the operation link. According to the analysis of the operation distribution area, most of the area in the open-pit mine stope is not the main working space, not subject to mechanical disturbance, or the frequency and degree of mechanical disturbance is very small; thus, it can be considered as a non-human disturbance area (referred to as non-disturbance area). Dust generation in the non-disturbance area is characterized by non-mechanical dusting, mostly natural wind dusting. Dust generation in open-pit mine operations involves the generation and diffusion of dust, which is the source and diffusion source of dust. The non-disturbance zone does not involve dust crushing generation, mainly from the settlement dust generated by other operation links. The small amount of dust from the atmosphere of the mining area is the secondary dust source, which is the main diffusion source of dust in the pit stope. In general, the open-pit mine non-disturbance zone can be further divided into the stope non-disturbance zone and dump non-disturbance zone. Figure 2 shows some of the main classifications.

### 2.2. Classification of Dust Suppressants

Coal mine pollution seriously threatens mine production safety and the occupational health of miners, and chemical dust suppression can effectively reduce the concentration of coal dust. The currently recognized dust suppressants can be divided into three types: wetting chemical dust suppressants, condensing chemical dust suppressants, and bonded chemical dust suppressants [29], this is shown in Figure 3.
(1)Wetting chemical dust suppressants

A wetting dust suppressant is composed of one or more surfactants and some inorganic salts, which can increase the wetting ability and dust suppression effect of water on dust, especially suitable for fine dust. The dust suppression principle of a wetting dust suppressant is to change the contact state between water and dust through the interfacial adsorption layer formed on the surface, and to maintain the moisture content of dust particles by absorbing water in the surrounding environment through inorganic salts, so as to improve the dust suppression time.
(2)Coalescing chemical dust suppressants

The main component of a condensation-type dust suppressant is an oily substance. In its external phase, water first comes into contact with dust, wets it, and then condenses the dust. In contrast, the free surfactant molecular water increasing group in the solution breaks through the air film on the surface of the dust particles and forms a “bridge” between the water and dust particles to pull the dust into the emulsion so that the dust particles form a larger combination state.
(3)Bonding chemical dust suppressants

Consolidation is the main dust suppression mechanism of bonded chemical dust suppressors. Through covering, silicification, bonding, and polymerization, after spraying the dust suppressant solution, it will form a curing layer with a certain strength and hardness on the dust suppression surface (usually the road surface or the surface of loose materials) to fix the dust particles, to achieve the effect of dust suppression.

The dust particles in the non-disturbance area of the mine are mainly dusted by natural wind disturbance. The dust suppression idea can be determined to use an adhesive dust suppressant to form a hardened surface that can resist natural disturbance on the dust surface of the non-disturbance area to prevent the generation of small dust particles; as long as the shell is not broken, it will not raise dust, and the dust pollution in the non-disturbance area of the open-pit coal mine is treated from the source as shown in Figure 4. In addition, the bonded chemical dust suppressant has an excellent consolidation effect. It has the characteristics of a long retention time and good economic benefits.

According to the wind statistics in 2019, the number of windy days of grade 3 and above in the Jungar mining area in Northern China (including the Heidaigou open-pit coal mine and Haerwusu open-pit coal mine) is 223 days, mainly concentrated in spring. In the Pingshuo mining area (including the Antaibao open-pit coal mine, Anjialing open-pit coal mine, and East open-pit coal mine), there are 129 windy days of grade 3 and above, mainly concentrated in spring; the Xinjiang Zhundong Mining Area (including Xinjiang Tianchi Energy South Open-Pit Mine, Xinjiang Tianchi Energy General Gobi No. 2 Open-Pit Coal Mine, Xinjiang Yihua Open-Pit Mine, Hongshaquan Open-Pit Coal Mine, Xinjiang Zhundong Open-Pit Coal Mine, etc.) had 103 windy days of grade 3 and above in 2019, mainly concentrated in spring and summer. It can be seen that the windy weather in several major mining areas is mainly concentrated in spring. Spring is the main period of reclamation, so the dumping site covering the topsoil without completing reclamation is the main source of dust pollution in the non-disturbance area and should be the key objective of dust prevention in the non-disturbance area.

### 2.3. Indoor Dust Suppressant Performance Test


(1)Preparation of dust suppressant


The mine dump area is large, and the economy is an essential indicator of dust reduction in the dumping site. Therefore, the main choices of dust suppressants are pregelatinized starch ((C_6_H_10_O_5_)_n_) and non-ionic polyacrylamide ([-CH-CH_2_-CH-CH_2_-]_n_CONH_2_), and the ratio of the two is 10:1. Seven different concentrations of dust suppressants were configured, with concentrations of 0.1%, 0.3%, 0.5%, 0.7%, 0.9%, 1.1%, and 1.3%, respectively. The viscosity of different concentrations of dust suppressant is measured at room temperature with a rotational viscometer, and the chemical reaction process between starch and polyacrylamide was shown in Figure 5 [30].
(2)Dust suppression rate test

Firstly, the sample in the non-disturbance zone is screened using a 2-mm-diameter grading sieve, and the sieved sample is dried in a constant-temperature drying oven for 24 h. The temperature is maintained at 105~110 °C, and the treated sample is used as a test sample. A total of 57 samples were designed, with 10 g of loess in each Petri dish, of which 1 was not treated. We sprayed different concentrations of dust suppressant in equal amounts into 8 samples.

After the dust suppressant was cured, the samples sprayed with different concentrations of dust suppressant were taken out every 24 h and placed in the air duct testing machine for the wind resistance test. The wind speed was 5 m/s, and the load cell quality loss data were recorded after 0.5 h, and they were recorded as *m*_1_. The control group was tested in the same environment, and the load cell mass loss data were recorded as *m*_2_. We calculated the dust suppression efficiency with the following the formula:(1)$$\eta =\frac{m_{2}-m_{1}}{m_{2}-10}$$
(3)Dust suppressant crusting test

We placed a quantitative yellow sand sample into a Petri dish, sprayed the dust suppressant on the yellow sand sample, obtained a sample after crusting, placed the sample in an incubator, and removed it every 1 h to test the strength of the crust. We placed a quantitative yellow sand sample into a Petri dish, sprayed the dust suppressant on the yellow sand sample, obtained a sample after crusting, removed it after storing it in an incubator for 12 h, and measured the thickness of the dust suppressant crust.
(4)Growth test of dust suppressant on plants

Considering that dust suppressant crusts may affect the normal growth of reclaimed vegetation, a test of the effect of dust suppressant crusts on plant growth was designed. The test content is as follows:The topsoil of the dumping site in the non-disturbed area was selected as the test soil.We select 4 garlic plants with similar health, planted them separately, and watered them in equal amounts.We chose 2 garlic plants to spray 1% and 0.5% dust suppressant at 100 mL, respectively. The rest did not receive dust suppressant. We only sprayed the same amount of water.We recorded the 3 garlic seedlings with the best growth conditions in each pot of test garlic over time, took the average value, and drew the growth curve of the garlic.
(5)Metal corrosivity test of dust suppressant

The non-disturbed area’s dust suppressant may cause corrosion to the loading equipment during transportation and spraying and cause corrosion to mining machinery after drifting or volatilization after spraying, so it is necessary to study whether this dust suppressant will affect the corrosion rate of the metal to avoid damage to mining equipment. We designed the experiment as follows.

In order to ensure the authenticity of the experiment as much as possible, the experimental specimen was selected from the same material as the mining equipment, as shown in Figure 6. All the specimens were submerged into the dust suppressant solution, fully soaked for 36 h, and weighed to calculate their quality.
(6)Dust suppressant toxicity test

An acute oral toxicity test, acute skin irritation test, and heavy metal element test for dust suppressants were performed, respectively.

1. Acute oral toxicity test with mice as the test object. First, the test sample should be dissolved or suspended in a suitable excipient, water, or vegetable oil as a solvent. When preparing a suspension, one cannot use organic chemical solvents with obvious toxicity. Secondly, healthy adult mice or rats are selected, and the difference in weight between individuals of the same sex should not exceed 20% of the average body weight. Before the test, the animals should adapt to the test environment for at least 3 d~5 d. Finally, according to the requirements of the selected method, in principle, 4~5 dose groups should be set up, and each group of animals is generally 10, half male and female. The spacing between each dose group should consider the toxicity and death. The pre-test is usually carried out with a larger group distance and fewer animals. If the toxicity of the test sample is very low, the maximum limit method can also be used, i.e., 20 animals (male and female), using a dose of 10,055.2 mg/kg. If no animal death is caused, multiple doses of acute oral toxicity tests can no longer be performed.

2. The acute skin irritation test took rabbits as experimental subjects and removed the coat with an area of 3 cm × 3 cm on both sides of the animal’s spine 24 h before the test. A 1 mL sample of dust suppressant solution was applied to the dehaired skin of animals. The degree of stimulation of the subject’s skin to the dust suppressant sample after 24 h, 48 h, and 72 h was recorded.

3. The heavy metal content test adopts atomic fluorescence spectroscopy (XRF) and inductively coupled plasma mass spectrometry (ICP-MS). Atomic fluorescence spectroscopy (XRF) first prepares a solution, adding nitric acid acidification at 1% to the water sample of the dust suppressant for storage. We prepare 3 sets of samples with the same conditions, take the standard solution (dust suppressant standard solution 100 μg/L) at 0, 1, 3, 5, 7, 10 mL in a 100 mL volumetric flask, and add 5 mL of concentrated hydrochloric acid, 5 mL of 10% thiourea solution, and 5 mL of 10% ascorbic acid solution to the sample and series of standard solutions, mix well, and set the volume to 100 mL. We calculate the recovery rate, open the argon valve, adjust the partial pressure gauge to 0.2~0.3 Mpa, turn on the computer and instrument host and network, preheat for 30 min, detect the current carrier and reducing agent, and, after the instrument is stable, we start to measure the fluorescence intensity of the standard solution and the sample, and calculate the content of heavy metals in the sample after obtaining the standard curve. Inductively coupled plasma mass spectrometry (ICP-MS) first takes the prepared 1 μg/mL mixed standard solution. We formulate a series of standard solutions of 0, 0.5, 1, 2, 5, 10, 20, and 50 ng/mL. Weigh three 0.l g (0.000l g accurate) dust suppressant samples in parallel and add them to a 100 mL PTFE beaker. We add 2 mL of nitric acid and 1 mL of hydrofluoric acid, respectively, and heat them on an electric furnace to dissolve. When the reaction is complete, the sample is cooled for 30 min and we transfer the solution to a 100 mL volumetric flask. We edd 0.1 mL and 1 mL of 1 μg/mL of mixed standard solution to two of these volumetric flasks, respectively. We add 1 mL of 1 μg/mL Sc, Y, In, Cs to mix the internal standard solution and set the volume. This was used to calculate the recovery rate of each element, and three spiked recovery experiments were carried out in parallel.

### 2.4. On-Site Dust Suppression Efficiency Test

The preparation of dust suppressants for the field test is divided into three links: dust suppressant preparation, reagent loading, and test site spraying. We pour 1.5 t of water into the dust suppressant solution preparation container, and then sprinkle the proportionally configured original dust suppressant powder into the bucket, where the ratio of dust suppressant powder to water is 1:100; that is, 10 kg of dust suppressant is added per 1 t of water. The dust suppression material was added to the water 5 times and stirred after standing for 3 h to obtain a dust suppressant. Field tests were conducted in windless weather, dust suppression efficiency tests were carried out using fans and dust concentration sensors, and site maps were taken 5 days and 1 month after the dust suppressant was sprayed.

Since this test was conducted on-site, gasoline-type fans were selected for wind load disturbance, considering portability and mine energy supply factors. We placed an anemometer at the outlet of the fan and recorded the wind speed together with the dust tester. Among them, the fan is selected YT9502 gasoline fan produced by China Shandong Linxing Power Machinery Co., Ltd. (Linyi, China), the dust detection is selected from the model SDL307 detector produced by China Qingdao Lubo Weiye Environmental Protection Technology Co., Ltd. (Qingdao, China), and the wind speed detection is produced by China Shanghai Bangwo Instrument Equipment Co., Ltd. (Shanghai, China). The model is AS836 Seema anemometer.

Through the above equipment, real-time dust data and wind speed data can be obtained during the working process of the fan. The monitoring range of PM2.5 and PM10 of the dust concentration detector is 0–2000 mg/m^3^.

## 3. Results and Discussion

### 3.1. Laboratory Test Results

#### 3.1.1. Dust Particle Size Distribution

To further analyze the dust particle size distribution in the disturbance zone and non-disturbance zone, on-site dust collection was carried out, and the particle size distribution of dust was tested, and the results are shown in Table 1. Among them, D10, D50, and D90 indicate that the sample particle size accounting for 10%, 50%, and 90% of the total particle volume is lower than this value, and D50 is also considered as the median particle size. It can be seen from Table 2 and Table 3 that the particle size distribution of dust 1# for three tests in the non-disturbance zone is similar. The median particle size is approximately 21 μm, the dust particle distribution below 80 μm is the greatest, and the dust below 1 μm is the least. The particle size distribution of the dust 2# sample in the disturbance area also has good similarity. The median particle size is approximately 36 μm, the dust particle distribution below 230 μm is the greatest, and the dust below 1 μm is the least, which is consistent with the above distribution characteristic map analysis. From the D50 and D90 data values of the 1# and 2# dust samples, it can be found that the median particle size of 2# dust is significantly greater than the median particle size of 1# dust. The D90 of 2# dust is much larger than the D90 of 1# dust, and the contrast is more obvious. It can be concluded that the overall dust particle size in the disturbance zone is larger than the dust particle size in the non-disturbance zone. This is because most dust particles in the non-disturbance area originate from the disturbance area. The dust particles generated by mechanical operations in the disturbance area are larger, directly settling in the area, but some of them are transformed into smaller dust particles. The smaller dust particles are lifted by the wind and brought to the non-disturbance area without interference. Thus, the size of the dust particles in the non-disturbance area is generally smaller than in the disturbance area.

#### 3.1.2. Dust Suppressant Efficiency Analysis

We placed a quantitative yellow sand sample in a Petri dish, sprayed the dust suppressant on the yellow sand sample, obtained a sample after crusting, placed the sample in an incubator for storage, and took out the sample every 12 h and placed it into a hair dryer testing machine to detect the weight loss of the sample. The experimental results are shown in Figure 7.

Figure 7 shows that the dust suppression efficiency of the dust suppressant solution will increase with the increase in the quality of modified starch in the dust suppressant. The dust suppression efficiency increases rapidly when the concentration of the dust suppressant is 0.1~0.7%, because starch can act as a water absorbent and flocculant in the dust suppressant to improve the dust suppression efficiency. The dust suppression efficiency of the dust suppressant will decay with the increase in time, the minimum attenuation is 10%, and the dust suppression effect of each concentration from the best dust suppression agent will be attenuated to 10% within 3 d because, with the increase in time, the evaporation of water in the dust suppressant weakens the dust suppression effect of the dust suppressant. When the concentration of dust suppressant is above 0.7, the dust suppression efficiency within 12 h is more than 90%, of which the dust suppression effect is maintained at 90% when the concentration of dust suppressant is 0.7. The time is less than 4 d, and the holding time of the dust suppression effect increases as the concentration of the dust suppressant continues to increase. When the dust suppression concentration is 1.3, the dust suppression effect of more than 90% of the sample dust suppression effect retention time reaches 6 d. In order to solve the problem of severe dust pollution in the process of open-pit coal mining and filling, Zhao et al. [31] proposed a dust suppression technology and determined the best mass–concentration ratio through the orthogonal test. The results of the wind tunnel simulation test of the dust suppressant configured with the optimal concentration ratio show that the suppression efficiency of the composite dust suppressant on the total dust and inhalable dust can reach 81.90% and 87.06%, respectively, at a wind speed of 4.00 m/s. The dust suppressant can effectively improve the environmental quality of open-pit mining.

#### 3.1.3. Analysis of Dust Suppressant Characteristics

Figure 6 shows the viscosity, crusting time, and crusting thickness of different concentrations of dust suppressants.
(1)Viscosity of dust suppressant

The viscosity value of the dust suppressant solution increases with the increased quality of the modified starch in the dust suppressant; see Figure 6. Field applications require a certain amount of modified starch to obtain a sufficient dust suppression effect and also require dust suppressants to have a certain fluidity to meet the on-site spraying. The viscosity range of the dust suppressant is 0.182~0.25 Pa·s—that is, the concentration range of the dust suppressant should be 0.7~1.2%.
(2)Dust suppressant crusting time

Figure 6 shows that the crusting time of the dust suppressant in the concentration range of 0.1~0.7% increases with the increase in dust suppressant. The crusting time reaches 29 h when the concentration is 0.7%. As the concentration of dust suppressant continues to increase after 0.7%, the crusting time shows a trend of decreasing with the increase in concentration. As the concentration increases, the viscosity of the dust suppressant increases, and the concentration of the dust suppressant increases, resulting in a decrease in the flow degree of the solution, and it cannot penetrate the sample, resulting in a thin crusting thickness, thereby reducing the crusting time.
(3)Thickness of dust suppressant crust

It can be seen from Figure 8 that the thickness of the crust of the dust suppressant in the concentration range of 0.1~0.9% will increase with the increase in the dust suppressant. The crusting time reaches approximately 10 mm when the concentration is 0.9%. As the concentration of the dust suppressant continues to increase, the crusting thickness decreases with the increase in concentration after 0.9%. This result may be related to the viscosity of the dust suppressant, which increases the concentration of the dust suppressant and causes the solution flow to decrease, and it cannot penetrate the sample, resulting in a thin crust thickness.

According to the above tests, considering the economic cost, dust suppression efficiency, and application time, a 1.0% concentration of dust suppressant (0.9% pregelatinized starch and 0.09% non-ionic polyacrylamide mixture) was selected as the final dust suppressant product and applied to the field for testing.

#### 3.1.4. Effect of Dust Suppressant on Vegetation Growth

Since the dust suppressant is a crust-type chemical dust suppressant, its dust suppression mechanism is to cement solid microparticles to each other through the dust suppressant as the cementing agent and form a consolidated layer on the surface, with the dust suppressant as the skeleton and the solid microparticles as the main body. Considering that this phenomenon may affect the normal growth of plant seeds, this experiment was designed to explore the effect of dust suppressant crusting on plant growth. This test used garlic as the test material for the test, and the test content was as follows:Sampling in the non-disturbed area.Select 4 garlic plants with similar health, plant them separately, and water them in equal amounts.Configure 100 mL of 1% concentration dust suppressant, and select 2 pots of test garlic for watering and marking.Record the 3 garlic seedlings with the best growth conditions in each pot of tested garlic over time, take the average value, and draw the growth curve of the garlic.

The curve drawn according to the daily measurement of garlic seedling length is shown. We chose garlic as the test plant because the duration of the dust suppressant’s harmful effect on the plant is very short. With the use of a longer growth cycle of the plant, the dust suppressant loses its effect, the plant does not grow, and we cannot obtain the test results, so we chose garlic, with a short growth cycle in a short time, as the garlic growth rate is fast. The experimental results are obvious and are convenient for test comparison.

According to Figure 9, garlic without dust suppressant germinated earlier than garlic with dust suppressant. The garlic seedlings began to germinate on the 4th day, when the concentration of dust suppressant increased to 1.0%. In addition, the growth rate of the three groups of garlic seedlings was basically the same after emergence, which may be due to the fact that the layer hardening shell formed on the surface of the loess after spraying the dust suppressant hindered the germination time of the garlic seedlings. The dust suppressant did not affect the growth state of garlic seedlings, which remained basically the same, proving that the addition of the dust suppressant did not affect the growing health of the garlic seedlings. In order to solve the problem of road dust pollution, Du, CF [32] used monomer, orthogonal, and optimization experiments and developed ecological dust suppressants for pavements, including moisturizers, moisture absorbers, coagulants, and surfactants, based on the dust mechanism. The preliminary formula of the dust suppressant was obtained through orthogonal experiments, and the toxicity to plants, moisture absorption and retention, and relative damage rate to plant seeds were used as experimental indicators. The experimental results show that the ecological and environmentally friendly dust suppressant displays good moisture absorption and moisturizing performance, good resistance to wind and rain erosion, no effect on plant growth, and no toxicity. At the same time, it can also bring good economic and social benefits, showing its broad application prospects.

#### 3.1.5. Effect of Dust Suppressants on Gold Corrosion

To ensure the authenticity of the experiment as much as possible, the experimental specimen is selected from the metal iron sheet of the same material as the mining equipment. All the specimens are submerged into the dust suppressant solution, fully immersed for 36 h, and weighed to calculate the quality. Record the experimental results as shown in Table 2. 

**Table 2 ijerph-20-00934-t002:** Metal corrosion detection.

Test Project	Test Method	Unit	Test Results
Total Penetration Corrosion (Cast Iron)	GJB5974-2007	mg/(cm^2^·24 h)	0.039
/	1#	2#	3#
Cast iron is dried and weighed (before corrosion)	19.864 g	20.031 g	19.903 g
Cast iron is wiped and weighed (after corrosion)	19.860 g	20.019 g	19.895 g
Poor quality	0.004 g	0.013 g	0.008 g

According to the test results, the total corrosion rate of cast iron is 0.039 mg/(cm^2^∙24 h). Therefore, the annual corrosion of cast iron by the dust suppressant is
0.039 × 365 = 14.235 mg/(cm^2^·a) = 0.014235 g/(cm^2^·a) 

According to the data review, the density of cast iron is approximately 7.4 g/cm^3^, and the annual corrosion rate can be obtained:0.014235 g/(cm^2^·a) ÷ 7.4 g/cm^3^ = 0.0019236 cm/a

The corrosion rate of cast iron is 0.0019236 cm/a—that is, 0.019236 mm/a.

According to the “Metal Corrosion Protection Manual”, on the corrosion resistance of metal materials, it can be seen that cast iron is very resistant to corrosion under the condition of dust suppressants. According to the data, the corrosion rate of cast iron in underwater conditions is 0.6~1.3mm/a. Its corrosion resistance is between corrosion resistance and under corrosion. The corrosion resistance of cast iron is reduced under dust suppressant conditions. Huang et al. [33] conducted a physical and chemical analysis of a new chemical dust suppressant for road dust in an open-pit mine. It was prepared with sodium polyacrylate as the binder, sodium carbonate as a hygroscopic agent, polyethylene glycol as a water retention agent, and alkyl glycoside as a surfactant. The physical and chemical properties and dust suppression properties of the dust suppressant were tested. The results showed that the pH value of the dust suppressant was 11.03, and the viscosity of the center points was 18.5 mPa. The surface tension was 28.1 mN/m, which was in a state of under corrosion for metal corrosion. Under the same conditions, dust suppressants are much more efficient than water for all types of dust and respirable dust and are suitable for road dust protection.

#### 3.1.6. Dust Suppressant Toxicity Test


(1)Acute oral toxicity test


The acute oral toxicity test is the first stage of the toxicology test and the most basic test item. The reason for the acute oral toxicity test is to provide a reference basis for the acute toxicity grading and labeling management of the product, to provide information on the health hazards induced by the tested product after oral contact in a short period of time, and to provide a basis for toxicity effects and dose selection for future toxicity tests such as subacute (slow). The acute oral toxicity test also has important reference value for the preliminary estimation of the target organs and possible toxic mechanisms of action. The acute oral toxicity test is performed by the Inspection and Quarantine Technology Center of Ningbo Entry-Exit Inspection and Quarantine Bureau.

This experiment takes ICR mice as the experimental object for the acute oral test. The test method is the maximum dose method; the mice are fasted for 4 h before the test, and the mice are infected through oral gavage. The gavage dose is 10,055.2 mg/kg, and the health status of mice within 14 days is continuously recorded. The test results are shown in Table 3 below.

**Table 3 ijerph-20-00934-t003:** Acute oral toxicity test results.

Number of Animals/Animal	Body Weight $\bar{X}$ ± SD/g				Number of Deaths/Only	Mortality/%
	0/a	7/a	14/a	14 Days of Weight Gain		
10	19.80 ± 1.19	24.2 ± 1.03	28.90 ± 1.16	8.90 ± 0.340	0	0
10	19.50 ± 14.70	23.0 ± 1.35	26.30 ± 1.73	6.80 ± 0.680	0	0

The experimental results showed that the mice experienced no poisoning for 14 days and were in normal health. According to GB 15193.3-2014 Food Safety Toxicological Evaluation Procedures and Methods, the sample belongs to the actual non-toxic grade.
(2)Acute skin irritation test

The acute skin irritation test was designed to test whether the dust suppressant had an irritating effect on the skin of mammals: with rabbits as the experimental subjects, the coat covering an area of 3 cm × 3 cm on both sides of the animal’s spine was removed 24 h before the test. A 1 mL sample of dust suppressant solution was applied to the dehaired skin of animals. The degree of stimulation of the subject’s skin to the dust suppressant sample after 24 h, 48 h, and 72 h was recorded. The test results are shown in Table 4 below.

The test results showed that the highest response score of skin irritation of the test subjects at each recorded time point during the test was 0. The sample was not irritating to the skin of mammals according to the classification of skin stimulation intensity.
(3)Heavy metal element test

We aimed to study whether the non-disturbance zone bonded dust suppressant developed in this paper contains heavy metal elements such as As, Cd, Cr, Hg, Pb, etc., which are potentially harmful to the human body and ecological environment. In this experiment, the JC-YZYG-300 atomic fluorescence photometer produced by Qingdao Jingcheng Instrument and Meter Co., Ltd. (Qingdao, China) was used under the condition of temperature of 18 °C and humidity of 48%, and the heavy metal content of the bonded dust suppressant in the non-disturbance zone was monitored by iCAP Qc inductively coupled plasma mass spectrometer produced by Thermo Fisher Scientific in the United States under the condition of temperature of 20 °C and humidity of 42%, and the test results are as follows.

It can be seen from Table 5 that the content of toxic elements is below the standard value, and it can be considered that the dust suppressant has no harm to the ecological environment and the human body. Li et al. [34] comprehensively considered the performance, environmental safety, and cost-effectiveness of chemical dust suppressants and constructed a comprehensive evaluation index system for chemical dust suppressant performance, including wetting performance, moisture absorption performance, adhesion performance, the annual cost per unit area, the pH value of the dust suppression liquid, chemical toxicity, etc. Among them, the toxicity of the dust suppressant was evaluated by its LD50. The indicator that characterizes the corrosion performance of dust suppressants is the corrosion rate of Q235 steel. The article also provides the classification of chemical acute toxicity, the corrosion rate of Q235 steel, and the corresponding standard test methods. The results are consistent with the phenomena observed in the experiment, in line with conventional understanding, and have strong confidence. This method can provide a standardized evaluation technique for the comprehensive performance evaluation and comparison of dust suppressants.

### 3.2. Field Test Results

#### 3.2.1. Dust Suppression Efficiency

The part of the loess area to be reclaimed in the dumping site was selected for field experiments. The selected area was divided into four areas. We sprayed the same amount of clean water and 0.7%, 1.0%, and 1.3% dust suppressants, respectively.

Figure 10 shows the dust suppression efficiency after crusting with different dust suppressants. Comparing the dust data of 0.7%, 1.0%, and 1.3% concentrations with those when no dust suppressant is sprayed, the dust suppression efficiency after the dust suppressant crusting can be clearly obtained, which can effectively solve the dust pollution of the Haerwusu open-pit mine. When the concentration of dust suppressant was 0.7%, the dust in the study area began to rise when the wind speed reached 5 m/s. The particle concentration of PM10 and PM2.5 increased rapidly from 0 mg/m^3^ to 1550 mg/m^3^ and gradually decreased when the wind speed exceeded 10 m/s. At this time, the dust suppressant crust is broken, but it still exists in the form of fragments and is not affected by the fan and flies. When the concentration of dust suppressant is 1.0%, the dust suppressant crust is broken at a wind speed of 20 m/s, the particles in the lower part of the crust rise, and the PM10 concentration is 1400 mg/m^3^. When the concentration of dust suppressant is 1.3%, the wind speed starts to blow dust from 5 m/s, peaks at 20 m/s, and reaches a peak PM10 particle concentration of 900 mg/m^3^. The dust suppression at a concentration of 1.0% and 1.3% is similar. Both have good dust suppression effects, but the cost is lower than that of a 1.0% concentration dust suppressant, and it is easier to use widely.

#### 3.2.2. Field Crusting Characteristics

Figure 11 shows the loss before and after spraying the dust suppressant. The loess without the dust suppressant is lumpy and loose, and the loess after spraying the dust suppressant is crust-like.

It was carried out in the soil dumping field in Haerwusu, and the test object was undisturbed to obtain scattered loess. The dust suppression effects of reagent crusts before and after dust suppressant spraying, before and after precipitation, and under long-term exposure were recorded. The spray volume was 0.67 kg/m^2^.

After spraying, the dust suppressant is obviously crusted on its surface, which can inhibit the flying of dust particles in its lower part. The shell of the dust suppressant is easily broken, but only partially, and the shell is intact as a whole. The loess without dust suppressant is not bonded between the particles, which is highly susceptible to stope dust. After destroying the loess, the surrounding loess slides towards the destruction pit.

The loess retaining wall shows a crusting phenomenon after spraying the dust suppressant. The average thickness of the shell after crusting is measured as 3.5 mm. The local dust suppressant accumulates, and the thickness of the shell exceeds 5 mm. The dust suppressant housing remains intact after breakage.

The test was carried out during a season of heavy precipitation in Xuejiawan Town, Ordos City, Inner Mongolia. After 5 days of spraying, the Haerwusu open-pit mine was subjected to light rain, with 8 mm of precipitation in 24 h. After the road in the mining area was unsealed, we observed the effect of precipitation on the dust suppressant after crusting. The loess sprayed with the dust suppressant was less affected by precipitation. It was found that the shell re-dissolves during precipitation and before and after precipitation, the dust suppressant recondenses after clearing, and the shell can remain intact. This phenomenon is in stark contrast to the erosion of loess retaining walls without dust suppressants. Dust suppressants can be affected by precipitation and have a long service life.

The state of the dust suppressant in the sprayed area was observed again after one month of spraying. During January, there was occasional precipitation in the test area, but the amount of precipitation was small. The surface of the dust suppressant crust was wet and wrinkled after prolonged exposure, but the shell was intact as a whole. The strength of the shell was reduced to a certain extent, the average thickness was only 1~1.5 mm, and it could still display a certain dust suppression effect after testing.

## 4. Conclusions

Open-pit coal mine dust pollution has become a prominent social problem. Especially in ecologically fragile areas, high-emission dust has seriously exceeded the environmental capacity, threatening ecological health. Taking the prevention and control of open-pit coal mine dust as the research background, this paper divides open-pit coal mine dust into the non-disturbance area and disturbance area according to the difference in dust generation. It develops an environmentally friendly, efficient, and economical dust suppressant in the dump employing a dust suppressant efficiency test, a test of the dust suppressant’s effect on plant growth, a toxicity test, and a field test. The main conclusions reached are as follows.

The dust starting power in the non-disturbance area is mainly natural wind flow, and the dust particle size in this area is smaller than the dust particle size in the disturbance area. The dust suppression efficiency of the dust suppressant synthesized by pregelatinized starch and polyacrylamide first increased and then decreased with the increase in the concentration of the dust suppressant. The dust suppression effect at a 1% dust suppressant concentration was the best. The use of dust suppressants will cause the growth of plants to lag but will not affect their normal conditions. The dust suppressant is non-toxic and harmless, and non-corrosive. The effective service cycle is more than 1 month, which fully meets the needs of dumping site reclamation. The best means of dust prevention and control in open-pit coal mines is source prevention and control, and dust reduction using dust suppressants is an effective means of dust source prevention and control. On the one hand, the use of a dust suppression agent in the dump can effectively reduce dust pollution, reduce the harm of dust in mining areas to the environment, climate, employees, and equipment, and, at the same time, reduce the water consumption required for dust reduction and aid in the ecological restoration of open-pit coal mines in ecologically fragile areas.

## Figures and Tables

**Figure 1 ijerph-20-00934-f001:**
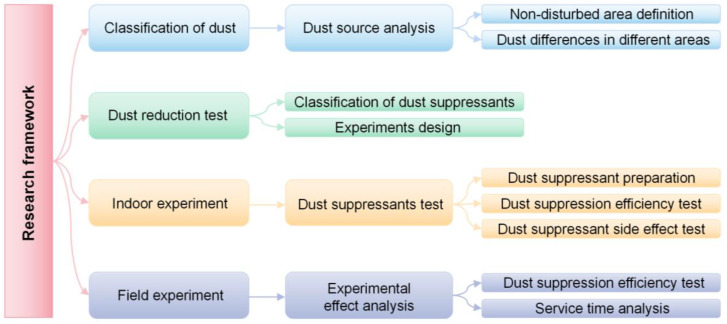
Research framework.

**Figure 2 ijerph-20-00934-f002:**
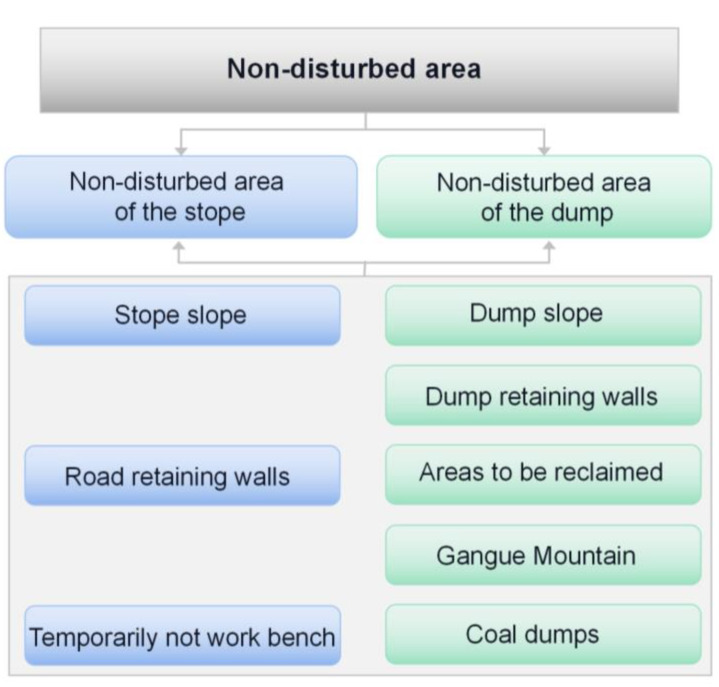
Non-disturbance zone of the open-pit coal mine.

**Figure 3 ijerph-20-00934-f003:**
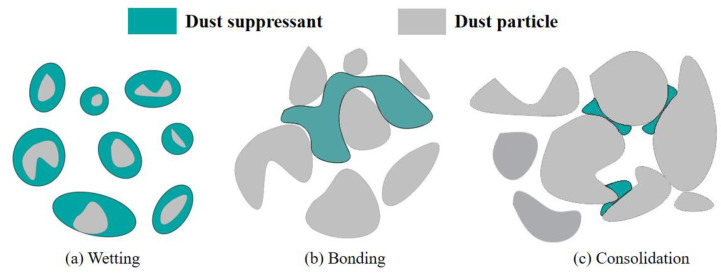
Dust suppression mechanism.

**Figure 4 ijerph-20-00934-f004:**
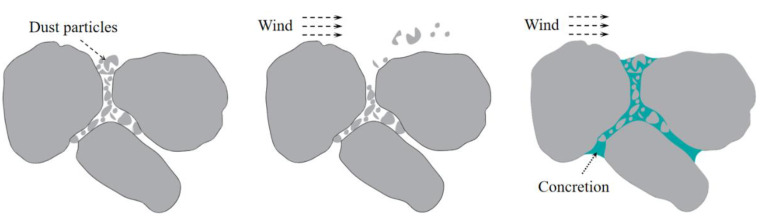
Principle of dust control in the non-disturbed area.

**Figure 5 ijerph-20-00934-f005:**
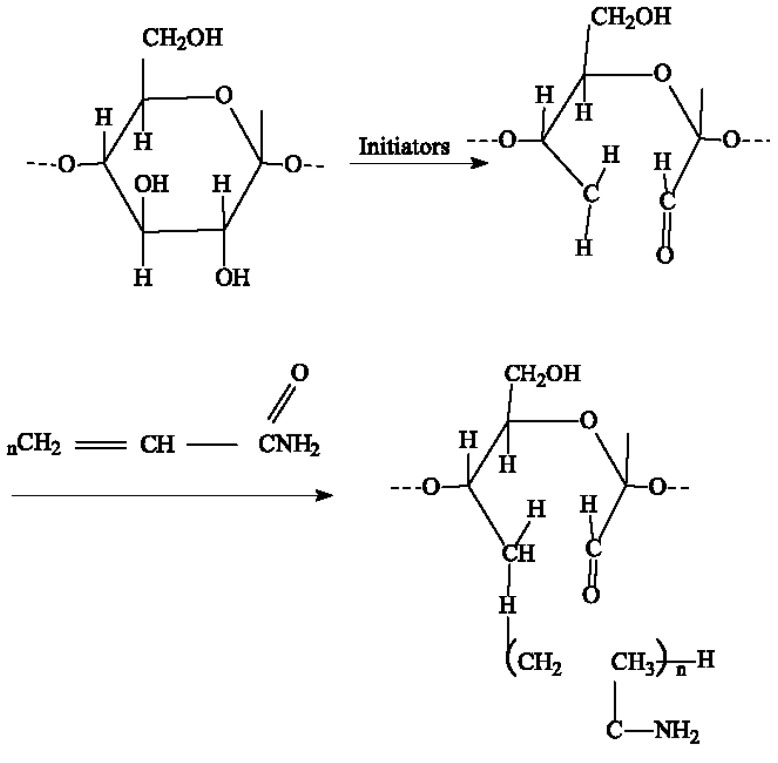
((C_6_H_10_O_5_)_n_) and ([-CH-CH_2_-CH-CH_2_-]_n_CONH_2_) reaction process.

**Figure 6 ijerph-20-00934-f006:**
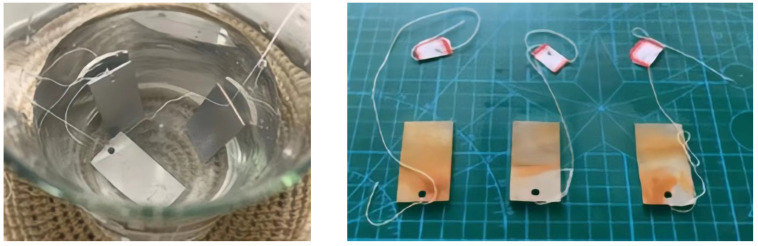
Metal corrosion diagram.

**Figure 7 ijerph-20-00934-f007:**
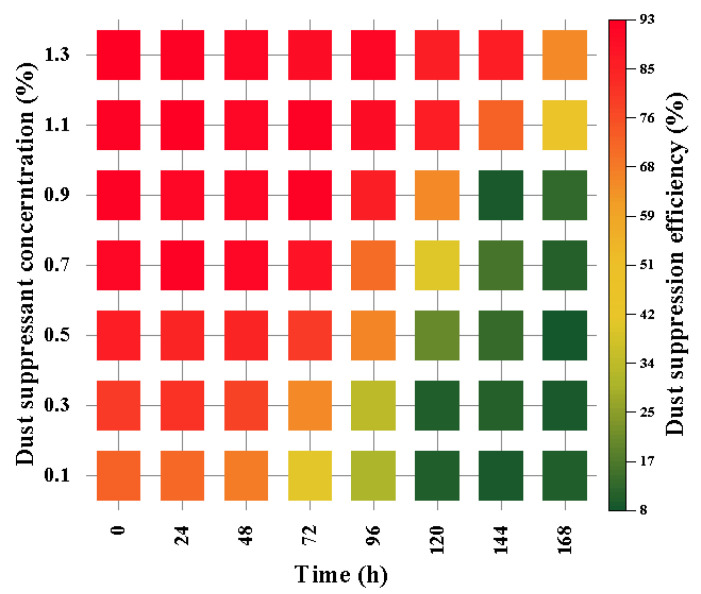
Dust suppression rate of dust suppressant at different times.

**Figure 8 ijerph-20-00934-f008:**
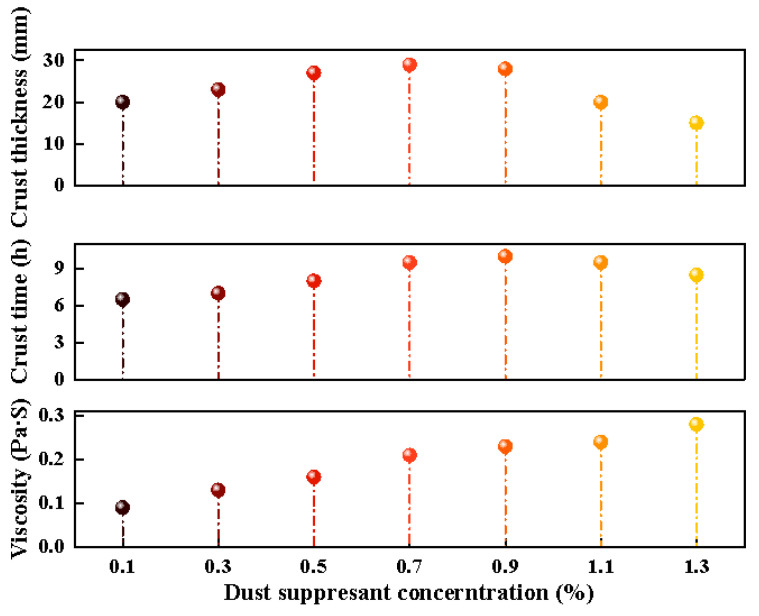
Viscosity, crusting time, and crusting thickness of different concentrations of dust suppressants.

**Figure 9 ijerph-20-00934-f009:**
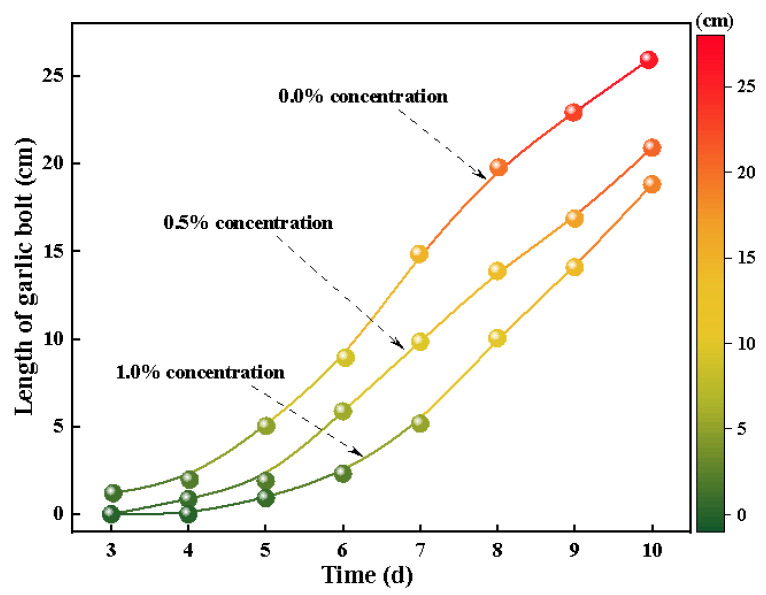
Length of garlic seedlings at different times.

**Figure 10 ijerph-20-00934-f010:**
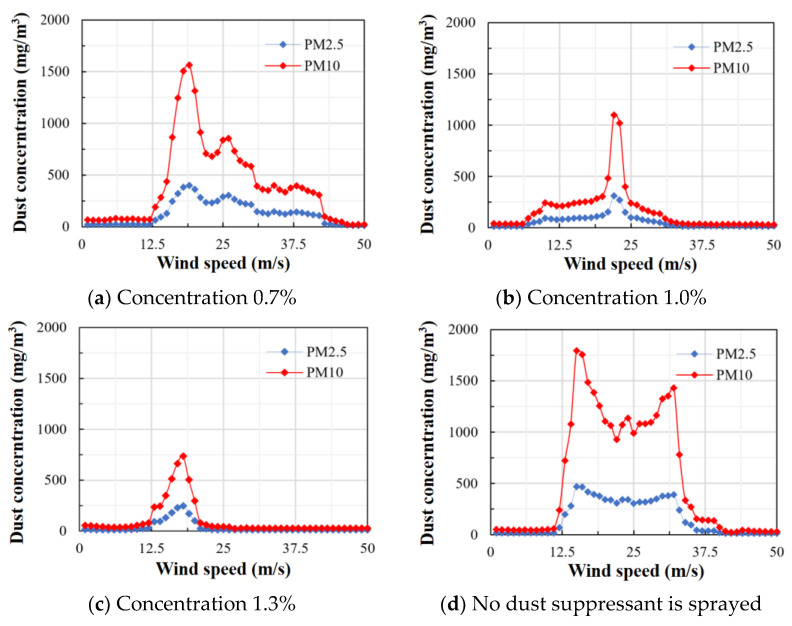
Dust concentration at different dust suppressant concentrations.

**Figure 11 ijerph-20-00934-f011:**
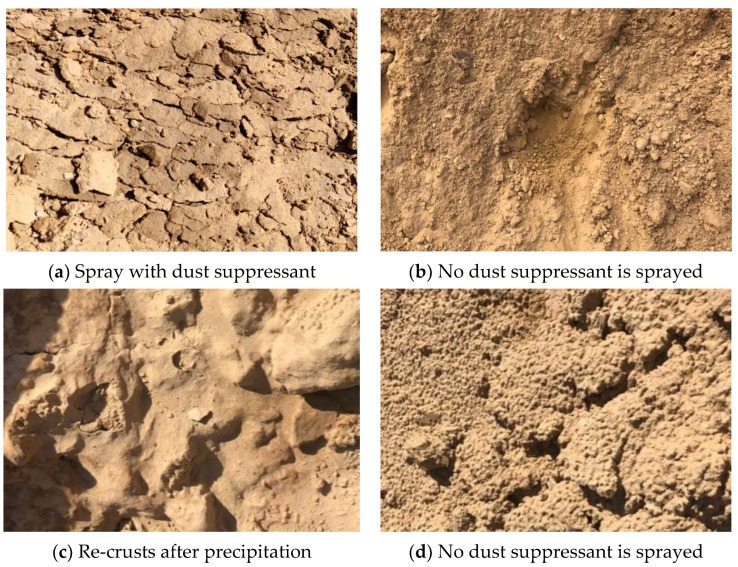
Comparison diagram before and after spraying dust inhibitor.

**Table 1 ijerph-20-00934-t001:** Particle size corresponding to dust volume distribution (μm).

		D10	D50	D90
Non-disturbed zone dust 1#	1st time	1.376	20.953	73.091
	2nd time	1.356	21.519	78.395
	3rd time	1.348	21.176	84.320
	average value	1.360	21.216	78.445
Disturbed zone dust 2#	1st time	1.835	34.888	239.770
	2nd time	2.080	39.928	229.74
	3rd time	1.804	32.973	210.260
	average value	1.902	35.888	226.130

**Table 4 ijerph-20-00934-t004:** Skin irritation responses.

Animal Number	Weight/kg	Observation Time					
		24 h		48 h		72 h	
		Sample	Control	Sample	Control	Sample	Control
1	2.44	0	0	0	0	0	0
2	2.18	0	0	0	0	0	0
3	2.72	0	0	0	0	0	0
4	2.63	0	0	0	0	0	0
Point mean		0	0	0	0	0	0

**Table 5 ijerph-20-00934-t005:** Test results.

Detection Item	As (mg/L)	Cd (mg/L)	Cr (mg/L)	Hg (mg/L)	Pb (mg/L)
Content	0.005	0.002	0.008	0.002	0.008

## Data Availability

Not applicable.

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
