# Peer review of "Study on Crust-Shaped Dust Suppressant in Non-Disturbance Area of Open-Pit Coal Mine—A Case Study"

_ijerph, 2023, doi:10.3390/ijerph20020934_

Round 1

Reviewer 1 Report

This study focused on " Dust Prevention and Control Strategy in the Non-Disturbance Area of Open-Pit Coal Mine - A Case Study" The original idea is good. The results could be interesting for other related researchers However, it’s hard to read and some corrections are needed before the final decision.

- The abstract needs a significant amount of reorganization to reflect the paper contents and the research contributions.

- The abstract didn’t show the data of the results. And didn’t show the control strategy of the dust in non-disturbance area.

- The key words cannot reflect the paper contents

- The introduction needs more highlights for the novelty and the weakness in the previous studies and should be re- organize to reflect the paper contents.

- ‘Materials and Methods’ should be re- organized to reflect how to do the experiment clearly. Please introduce the main components and chemical reaction process of the dust suppressant

- ‘0.039mg/cm2’ bad typesetting. Please check the whole manuscript.

- The manuscript didn’t introduce how to do the ‘Gold Corrosion’ experiment.

- The manuscript didn’t introduce how to do the toxicity test.

- The results should compared with other similar research.

- The title is about ‘Dust Prevention and Control Strategy’, but which part is analysis the control strategy?

- The ‘Conclusions’ is too complex and unattractive, please re-condense it and add the scientific value of your paper.

- Please cite more latest references.

- The figures and tables are too much. Pleas re-condense them to reflect the most important results.

Reviewer 2 Report

This paper investigated the performance of the dust suppressant, analyzed and tested five indicators of the dust suppressant through laboratory experiments, and completed the optimal performance ratio experiment. Overall, I find the paper well-structured and informative, yet with some details that deserved more attention. I would suggest that this study could be considered for publication after revisions as follows.

1.     The paper is in fact mainly dedicated to research on the dust suppressant, whether this could be highlighted in the title.

2.     Introduction line 57: The previous part is talking about the importance of dust control research, and why it turns directly to the suppression effect of spraying later. It is suggested to re-express it to make it a more logical relationship.

3.     Introduction line 105: Reasons for selecting this open-pit coal mine could be added to clarify the representativeness and necessity.

4.     Section 2.2. lines 154-158: Additional references are required for the classification and specific definition of dust suppressants.

5.     Section 2.3.: Does this part of the study refer to existing literature or is it a completely original innovation? If so, add the sources cited.

6.     Checking the unit, many places are not superscript. For example, lines 224 and 262.

7.     Please check the tables and try to keep the number of decimal places the same before and after. In addition, spaces shall be provided between numbers and units.

8.     Section 3.1.2.: In this section, only the results on efficiency are described. It is suggested that possible causes be added and further analyzed later.

9.     line 348: It is suggested to add reasons for the selection of garlic for the study.

10.  lines 381-381: Many details still need to be corrected.

Round 2

Reviewer 1 Report

Accept in present form